# Electrophysiological and Behavioral Responses of Virgin Female *Bactrocera tryoni* to Microbial Volatiles from Enterobacteriaceae

**DOI:** 10.3390/microorganisms11071643

**Published:** 2023-06-23

**Authors:** Anaïs K. Tallon, Lee-Anne Manning, Flore Mas

**Affiliations:** 1Department of Wildlife, Fisheries and Aquaculture, University of Mississippi State, Starkville, MS 39762, USA; 2The New Zealand Institute for Plant and Food Research Ltd., Canterbury Agriculture & Science Centre, 74 Gerald St, Lincoln 7608, New Zealand; lee-anne.manning@plantandfood.co.nz (L.-A.M.); flore.mas@plantandfood.co.nz (F.M.)

**Keywords:** *Bactrocera tryoni*, bacteria, microbial volatiles, electrophysiology, attractant, pest management

## Abstract

The Queensland fruit fly (*Bactrocera tryoni*) is a major polyphagous pest widespread in Australia and several Pacific Islands. Bacteria present on the host plant phyllosphere supply proteins, essential for egg development and female sexual maturity. We investigated the role of microbial volatile organic compounds (MVOCs) emitted by Enterobacteriaceae commonly found on the host plant and in the fly gut in attracting virgin females. Bacteria were cultured on artificial media and natural fruits, at various pH, and MVOCs were collected using different headspace volatile absorbent materials. The olfactory responses of virgin females to bacterial MVOCs were assessed via electrophysiology and behavioral assays. The production of MVOCs was strongly influenced qualitatively by the bacterial strain and the type of media, and it semi-quantitatively varied with pH and time. MVOCs emitted by *Klebsiella oxytoca* invoked the strongest antennal response and were the most attractive. Among the identified compounds triggering an olfactory response, D-limonene and 2-nonanone were both significantly behaviorally attractive, whereas phenol, nonanal, isoamyl alcohol, and some pyrazines appeared to be repulsive. This study deepens our understanding of the chemical ecology between fruit flies and their bacterial symbionts and paves the way for novel synthetic lures based on specifically MVOCs targeting virgin females.

## 1. Introduction

New Zealand is one of the last countries free of fruit flies (Tephritidae). However, its proximity to Australia, where the Queensland fruit fly (QFF), *Bactrocera tryoni* (Froggatt) (Diptera: Tephritidae) is widely present [1], represents a constant threat to New Zealand biosecurity. Originating from tropical and subtropical coastal Queensland, Australia, QFF has now spread to more temperate areas of southern Australia and established in several Pacific Islands and is regularly intercepted in New Zealand [1]. This fruit fly species is one of the most destructive crop pests worldwide due to great dispersal capacities, a generalist diet, and the ability to establish outside of its native range [2], which leads to considerable economic losses and draconian quarantine restrictions. Moreover, host plants of QFF include, but are not restricted to, apple and kiwifruit, which represent the main exported fruit crops of New Zealand [3]. Furthermore, the withdrawal of the organophosphate insecticides fenthion and dimethoate has left a major gap in pest suppression tools, and it has further increased the risk of incursion of QFF in the country. 

While male QFF can be controlled with strong attractants, for example, para-pheromones such as cue-lure (4-p-acetoxyphenyl-butan-2-one) or raspberry ketone (4-4-hydroxyphenyl-butan-2-one) [4], females remain relatively difficult to regulate in the wild [1,5]. Yet, female QFF, which sustain the population, are the ones that cause the most economic damage via egg laying in fruits [1,6]. Even though food-based attractants (e.g., fruit or protein-like yeast hydrolysates) can be mixed with insecticides and used in traps or sprays to target both immature sexes [7], these food baits display relatively low persistence. In addition, their efficacy is highly dependent on the size of the population and weather conditions [1]. As such, in the absence of a strong female-specific synthetic attractant, pest management alternatives are urgently needed.

Bacteria present on the host plant phyllosphere constitute the main source of proteins for female fruit flies [8], which is necessary for egg development and to reach sexual maturity [9,10,11,12]. Most bacteria identified from fruit fly digestive organs belong to the family Enterobacteriaceae, which constitute the dominant microbiome taxa associated with fruit fly infested fruits and plants [13,14,15,16]. Enterobacteriaceae, which are not obligate endosymbionts, are acquired by the insect from its environment while foraging for food on leaves and fruit surfaces [13,16]. Although previous chemical and behavioral studies have linked bacterial symbionts with attraction of fruit fly species, such as the Mediterranean fruit fly (*Ceratitis capitata*), the Mexican fruit fly (*Anastrepha ludens*), and the olive fruit fly (*Bactrocera oleae*) [17,18,19,20,21], the role of bacterial symbionts in the biology of *B. tryoni* remains poorly understood. 

Olfaction is essential for insects to locate potential food, mates, or search for suitable habitats, and oviposition sites [22]. Recent studies of plant–insect interactions have revealed that the attractiveness of the host plant to an insect is not only driven by the volatiles produced by the host plant but may also be influenced by microbial volatile organic compounds (MVOCs) present on the plant [22,23]. Yet, despite recent descriptive studies of the microbiome associated with *B. tryoni* [16,24], pest management efforts have been focused on using generic yeast hydrolysate or autolysate as sources of attractant, and not bacterial symbionts.

In this study, we investigated the olfactory responses of virgin QFF to four symbiotic bacteria strains belonging to the family Enterobacteriaceae, commonly found in association with QFF. MVOCs emitted from these bacteria were collected under different media and conditions using two different headspace volatile collection methods. Chemical profiles were identified through gas-chromatography coupled with mass-spectrometry (GC/MS), and responses of female QFF to MVOCs were assessed via electrophysiological and behavioral assays. 

## 2. Materials and Methods

### 2.1. Bacteria 

Four species belonging to the family Enterobacteriaceae were selected based on previously reported QFF gut bacteria [13]. In the absence of QFF in New Zealand, bacterial cultures were obtained from collaborators abroad and international collections. Among the family Enterobacteriaceae: *Citrobacter freundii* (CF) was provided by the South Australian Research Development Institute (South Australian Research and Development InstituteSARDI, Adelaide, Australia), *Enterobacter cloacae* (EC) and *Enterobacter* (syn. *Pantoea*) *agglomerans* (PA) were both obtained from the International Collection of Micro-Organisms maintained at Manaaki Whenua-Landcare Research (Lincoln, New Zealand), and *Klebsiella oxytoca* (KO) was imported from the Institut Agronomique néo-Calédonien (IAC, Port-Laguere, New Caledonia). In our experiments, only one strain of each bacterium was tested. Hereafter, we use their abbreviated names throughout the text, tables, and figures. 

### 2.2. Media

Artificial media commonly used with Enterobacteriaceae are tryptone soya agar (TSA) or tryptone soya broth (TSB), which were both sourced from Oxoid Basingstoke, Hampshire, England. Bacteria were first grown for 24 h on TSA at 29 °C ± 1 °C. Consequently, a loop of a colony was added to 100 mL of TSB in a 250 mL mason jar (Tablefair^®^, Auckland, New Zealand) and incubated on a rotary shaker at 30 °C and 100 rpm for 18 h. A standard concentration of 5 × 108 cells/mL (determined via spectrophotometry) was used for inoculation on broth media. The pH of cultures was measured using an electronic pH meter. The control TSB had a pH of 6.84, and the addition of bacteria increased the pH to 7.74. In addition, we tested the effect of pH on MVOCs production by the bacteria, as pH was previously reported to significantly influence the type of compounds released by food baits, and it consequently impacts the attraction of female QFF [18,25,26]. Given that the selected Enterobacteriaceae can be found both on host plants and within the gut of fruit fly (i.e., bacteria experience different pH conditions), the pH of the media containing the four inoculated bacteria was adjusted to 3, 5, 7, 9, and 12, using 12.4 M (38%) hydrochloric acid or 10 M (40%) sodium hydroxide solution in a dropwise manner until reaching the appropriate pH [26]. The pH was re-checked 24 h after adjustment and before the SPME collection commenced.

Fruits were also inoculated with bacteria to investigate the interaction between bacterium species and natural substrates. Mature green kiwifruit cv. ‘Hayward’ (*Actinidia deliciosa*), apples cv. ‘Royal Gala’ (*Malus domestica*), and oranges cv. ‘Navel’ (*Citrus sinensis*), purchased at the local supermarket, were surface sterilized by immersion in 85% ethanol for 2 min and then washed with distilled water for 2 min. Fruits were then left under UV light (100–400 nm) for 2 h, before being inoculated with CF, EC, KO, and PA. Each fruit was inoculated five times with 10 μL of the same bacterial solution (5 × 10^8^ cells/mL), 3 mm in depth, using a needle to mimic insect oviposition (Murphy et al. 1994). Control fruits were inoculated with 10 μL of distilled water only. Incubating conditions were the same as used for bacterial growth (i.e., 29 °C ± 1 °C). To confirm bacteria growth on fruits, a slice (ca. 1 cm²) of the flesh from the inoculation site was macerated with TSB and streaked onto TSA after the volatile collection and bacterial growth were assessed. 

### 2.3. MVOCs Collection and Identification 

Two different headspace volatile collection methods, using different absorbent materials, were performed to maximize the chance of capturing a maximum of MVOCs. Dynamic headspace collection was performed using an air pump to actively push/pull air through a sorbent (i.e., Tenax^®^-GR), which was solvent eluted to produce an extract that can be stored and later used for chemical analysis and bioassay. In contrast, passive headspace collection using solid phase micro-extraction (SPME) consisted of a fiber that passively adsorbs molecules and requires thermal desorption promptly after collection. SPME avoids solvent use, enables the detection of smaller volatiles that are generally hidden by the solvent peak, and generally allows a shorter collection time (minutes compared to hours). SPME was performed for 1 h to collect and compare MVOCs emitted from bacterial cultures grown on different media (artificial and fruits), and at different pH concentrations. In contrast, Tenax^®^ was used to collect bacterial headspace for 24 h, and the solvent-extracts were used later for electrophysiology and behavioral assays. All headspace collections were conducted in Mason jars with bacterial cultures growing either on TSB or fruits. For each bacterial culture, three replicates and a media control were collected via each aforementioned method. All collections were performed under BSL2 laboratory conditions for human risk pathogens at The New Zealand Institute for Plant and Food Research Limited (PFR) in Lincoln, New Zealand.

Tenax^®^ filters used for dynamic headspace collection were adsorbent filters made from 60 mg of Tenax^®^-GR 35/60 (Grace Davison Discovery Sciences, Epping, Australia) packed in a glass tube between two plugs of silane treated glass wool (Grace Davison Discovery Science, Chicago, IL, USA). Air was drawn in from the environment, passed through a charcoal filter, distributed over the culture, and adsorbed onto the Tenax^®^ filter at a rate of 500 mL/min. In between each collection, the charcoal filters were heated at 150 °C overnight while Tenax^®^ filters were conditioned at 250 °C for 3 h under a nitrogen flow. After 24 h of volatile collection, the Tenax^®^ filters were eluted with 1 mL of hexane (Sigma-Aldrich, Saint Louis, MO, USA, 95% purity). Two hundred microliters of each extract were concentrated down to circa 80 μL under a gentle flow of argon and used for chemical analysis. Pooled Tenax^®^ extracts of each bacterium were made by mixing 100 µL of each of the three replicates. A pooled control was made by mixing 100 µL of each fourth replicate from each bacterial control containing the media only (e.g., TSB). The pooled bacterial Tenax^®^ extracts were used for electroantennogram (EAG) and behavioral choice assays with QFF. All the bacterial Tenax^®^ extracts were from cultures grown at pH 7 (±0.5). Differences in the proportions of MVOCs emitted from kiwifruit, apples, and oranges, inoculated with either CF, EC, KO, or PA, were assessed at 24 h post-inoculation. Over time variations of chemical profiles, measured at 24 h, 72 h, and 5 days post-inoculation, were assessed for inoculated kiwifruit.

The SPME fiber selected in our experiments was a grey Stableflex/SS 2 cm SPME (50/30 μm DVB/CAR/PDMS, SUPELCO, Sigma-Aldrich, Saint Louis, MO, USA). The SPME fiber was directly inserted inside the Mason jar and exposed to the headspace of the culture a few centimeters above it for 1 h. Before each use, the SPME fiber was conditioned as recommended by the manufacturer (i.e., 270 °C for 30 min). MVOCs were collected from each bacterium at pH 3, 5, 7, 9, and 12, as well as mixtures of all bacteria adjusted to each pH, using SPME. Differences in chemical profiles of inoculated fruits with bacteria were also assessed with SPME at 24 h post-inoculation and compared with their Tenax^®^ counterparts. 

### 2.4. Chemical Analysis

Chemical analyses of bacterial Tenax^®^ extracts and SPME samples were performed on a gas chromatograph (GC, 7890B, Agilent Technologies, Santa Clara, CA, USA) coupled to a mass-spectrometer (MS, 5977A, Agilent Technologies, Santa Clara, CA, USA). The general GC operating conditions were split-less injections for 0.6 min using helium carrier gas at 1.6 mL/min. The GC column was non-polar (DB-5 ms, Agilent Technologies, Santa Clara, CA, USA) with the following dimensions: 30 m × 0.25 mm i.d. × 0.25 μm film thickness. The temperature program started at 40 °C (2 min hold) and was increased by 4 °C/min to 280 °C followed by a 10 min hold. The injector temperature for the column was 250 °C. The transfer line was maintained at 250 °C. The MS system was operated in electron impact mode at 70 eV. Automatic integration of each peak was performed with Masshunter software (Agilent Technologies, Santa Clara, CA, USA), which provided a list of compounds (i.e., peak) for each sample. Identification of compounds was then performed by matching the spectra of each peak with the NIST Mass Spectral Database library (Ver 2.2, 2014). In addition, Kovats retention indices (KI or RI) for each compound were calculated by running an external alkanes series using the aforementioned GC/MS method. Finally, identification of peaks-of-interest was confirmed with synthetic standards purchased from Sigma-Aldrich and marked as bold in Table 1. 

### 2.5. Queensland Fruit Fly (QFF)

Factory-reared *B. tryoni* pupae were provided by the Elizabeth Macarthur Agricultural Institute from the Department of Primary Industries (EMAI-DPI), located in Camden, New South Wales, Australia, and were used in electrophysiological (EAG) and behavioral (i.e., cage assay) experiments in Australia at Macquarie University Centre for Fruit fly Biosecurity innovation, Sydney. The same pupae were exported under quarantine to New Zealand with a permit from the Ministry of Primary Industries and approval from the Environmental Protection Authority to work on a new organism in containment (NOC100167). However, we had to use sterile insects in New Zealand. Therefore, pupae, which were provided by the EMAI-DPI, were first irradiated at 70–75 Gy using Gamma radiation (Cobalt-60) at the Australian Nuclear Science and Technology Organisation before being shipped to New Zealand. The irradiated insects were only used for electroantennogram detection (EAD) to identify bioactive compounds. Finally, complementary behavioral experiments (i.e., y-tube olfactometer) were carried out in New Caledonia with QFF reared at the Institut Agronomique néo-Calédonien (IAC), at the Station de Recherches Fruitières de Pocquereux, La Foa, New Caledonia, to test the identified bioactive compounds. All flies were raised under similar laboratory conditions, at 25 °C ± 1 °C and 70% ± 10% humidity and a photoperiod of 12 h light:12 h dark (light from 0600 to 1800 h) and a dusk and dawn of half an hour to induce mating behavior. Given that sexually immature females are expected to search actively for protein sources to mature and develop eggs, we used 3–4-day-old virgin females, fed only with water and sugar ad libitum [10].

### 2.6. Electrophysiological Assays

The electroantennogram (EAG) responses of virgin females to Tenax^®^ bacterial extracts were measured in Australia. In New Zealand, using GC coupled with an electroantennogram detector (GC/EAD), we separated and identified the bioactive MVOCs using SPME collection samples from the mixed bacteria grown under different pH concentrations. For both techniques (i.e., EAG and GC/EAD), each fly was anaesthetized at −20 °C for 2 min or by using carbon dioxide gas. The head was then removed and mounted between two recording electrodes in a salt-free hypoallergenic gel (Spectra 360, Parker Laboratories Inc., Fairfield, NJ, USA) and placed under a constant charcoal-filtered flow of humidified air (400 mL/min). For the EAG, the presentation of odorant samples was made manually in front of the antennae by inserting in the airflow a Pasteur pipette containing 10 µL of the bacterial Tenax^®^ extract applied on a filter paper and puffed for 0.5 s. Prior recording, the stimulus pipettes were kept for 10 min in a fume hood to facilitate solvent evaporation. For each sample, air was puffed (for 0.5 s at 1.5 L/min) through the Pasteur pipettes, which were inserted into a glass tube, located at 10 cm from the antennal preparation. A constant airflow was maintained to compensate for the air-puff. The samples were presented at 30 s intervals between stimuli to allow the olfactory receptor neurons to return to a basic activity rate as previously described [27]. The four bacterial Tenax^®^ extracts were tested in concentrated form, as well as in a series of dilutions (10-1 and 10-2) prepared and tested for a dose–response. All odorant stimuli were delivered from the lowest to the highest concentrations. Ten replicates were run for each bacterium and dilution. The EAG responses of female QFF to bacterial Tenax^®^ extracts were recorded with a high-input impedance preamplifier and analyzed with the recording software AutoSpike (version 3.9, Syntech, Hilversum, The Netherlands).

For GC/EAD, the SPME fiber passively collected MVOCs for 1 h at 29 °C before desorption in the GC injection port. The SPME samples were presented to the insect antennae via an Agilent GC equipped with a 30 m by 0.32 mm i.d. by 0.25 µm HP5 capillary column (Agilent Technologies, Santa Clara, CA, USA) using an FID coupled to an electro-antennographic detector (EAD) acquisition controller (IDAC 4, Syntech, Hilversum, The Netherlands). The fiber was desorbed for 5 min in the injection port, which was held at 250 °C. The samples were subsequently passed through the column at 1 mL/min with helium carrier gas, which was split between the insect antennae and the FID. The injector of the GC was set at 250 °C and run in split-less mode; the detector was set at 300 °C, and the GC oven temperature programmed from 60 °C, held for 1 min, then increased to 250 °C at 20 °C/min and held for 10 min. The EAD exit port temperature was maintained at 250 °C, and the antennal preparation was placed in a charcoal-filtered and humidified air system (400 mL/min). The recording software AutoSpike (Version 3.9, Syntech, Hilversum, The Netherlands) was used to record the antennal response and FID response. Kovats retention indexes (KI) were calculated for the antennally active compounds, and the samples were run by GC/MS for identification. 

### 2.7. Behavioral Assays

Cage assays were performed in Australia to first assess the attraction of virgin female QFF to the whole bacterial Tenax^®^ extracts. Experiments were run between 9 am and 4 pm in a controlled temperature room at 25 °C with 65–75% relative humidity with 18 fresh air changes every hour. Thirty females were set up in small mesh cages (30 × 30 × 30 cm) with sugar and water ad libitum for 24 h prior to the start of the experiment to allow acclimation. After 24 h, two 20 mL vials lined inside with Tanglefoot^®^ glue were hung from the ceiling of each cage on either side of the middle of the cage. Each vial contained a piece of filter paper (approx. 2.5 mm × 3 mm) with 10 µL of undiluted Tenax^®^ extract from either a bacterium or the control solvent (n-hexane, Sigma-Aldrich, 95% purity). The position of the bacterial extract in the cage was randomly assigned daily across all replicates between the right and the left side. The number of female QFF caught in each of these two traps was assessed after 24 h. Ten replicates of each of the four bacteria included 30 females per cage and were tested consecutively after four days, leading to a total of 1200 tested females.

Y-tube olfactometer assays were performed in New Caledonia with synthetic compounds selected based on identified bioactive compounds following the method described in [6]. One fly was introduced in the entrance of each olfactometer and allowed to acclimate for 5 min. One microliter (i.e., 1 mg) of a pure synthetic odorant was applied to a piece of filter paper (Whatman, Sigma-Aldrich, Darmstadt, Germany) 30 mm × 7.5 mm, placed inside a Pasteur pipette, and connected upwind to one end of the Y arm. On the other arm of the Y-tube, another Pasteur pipette was set up with filter paper containing no odor. The arm of the Y-tube that was used for the odor was alternated between each run. Each compound was tested in a set of five runs, giving a total of 20 virgin females tested for each compound. After acclimatation, the doors from the entrance were lifted, and flies were given 15 min to make a choice between both arms of the Y-tube. Flies were considered to have made a choice when they crossed the door of one of the end arms, and were manually enclosed in it. The side of their choice was recorded, as well as the time of their choice. If the fly had not made a choice by the end of the 15 min, this was recorded as no choice. This experiment was carried out in New Caledonia with laboratory-reared flies, as described above. Behavioral tests were performed from 8 am until 4 pm, under constant light and temperature conditions (25 °C +/− 1 °C). Olfactometers were rinsed at the end of every day with bleach and water and left to dry overnight.

### 2.8. Statistical Analysis 

All the statistical analyses were performed with R statistical software, version 4.2.2. [28]. From the GC/MS results, relative proportions of each compound identified with each method, emitted by each bacterium cultivated on the TSB medium, were calculated based on the peak area of each volatile. The peak area of individual compounds was identified and recorded from the GC trace in one sample and divided by the total area of all compounds identified in the sample (see Appendix A for representative GC traces). Overall scent composition of samples was compared between the two methods (i.e., Tenax^®^ vs. SPME) by inspecting the relative proportion results reported in tables. We used non-metric multidimensional scaling (NMDS) and analysis of similarity (ANOSIM), from the R package named vegan [29], to determine variations in the MVOC profiles of the samples. Both qualitative (presence/absence) and semi-quantitative differences in the chemical profiles were evaluated among the four bacteria strains, types of fruits, post-inoculation time, and pH. The SPME-GC/EAD responses to the mix of the four bacteria were compared among the identified bioactive compounds and at different pH with linear models including bacteria and pH as fixed effects along with their interaction. 

The mean intensity and the standard deviation of EAG-responses were calculated and presented for each bacterial extract. Because the antennal sensitivity may decline during each recording session, each female was taken as a random factor in a linear-mixed effect model (LME) using packages “lme4” and “lmer” to control for variation between each female antennae replicate. The bacterial extract and the concentration, as well as their interaction, were included as fixed effects in the model. Because no effect of the concentration, nor interaction between the concentration and the bacteria strain were found, only the effect of bacteria was measured. We used Tukey-HSD to run pairwise comparisons between the bacteria. 

For both behavioral assays (i.e., cage and Y-tube), we compared the total number of females making a choice (odor or control) vs. the number of females making no-choice. Among the females making a choice, we then compared the number of females choosing the odor (bacterium/synthetic compound) vs. the control odor (solvent/blank). A general linear model (GLM) with a binomial family and a log-link function was used. The number of females was chosen as potential explanatory response variable, the tested odor was considered a fixed effect, and the total number of females making a choice was used to weight the model. Multiple comparisons between the odors were performed with least-square means contrasts (package “lsmeans”). The package “ggplot2” was used to draw the plots.

## 3. Results

### 3.1. Chemical Identification and Semi-Quantification

#### 3.1.1. Comparison of Headspace Methods

To investigate how the use of various odor collection methods can affect our representation of odor profiles, we identified and compared the chemical composition of headspace volatile collection between the Tenax^®^ filter and SPME fiber (Table 1; Appendix A). Variations in the overall chemical composition of the tested samples were assessed using NMDS and ANOSIM, which revealed a clear association of certain compounds to each of the two methods (Table 2). Altogether, Tenax^®^ headspace extracts collected a wider range of molecules compared to the SPME technique, which may be explained by a longer time of collection. In contrast, the SPME technique detected shorter hydrocarbon-chain volatiles such as 2-butanone, isobutyl-alcohol, acetoin, and other compounds likely concealed by the solvent peak in Tenax^®^ extracts. With both headspace collection methods, several pyrazines were detected in all bacteria, as well as from the TSB media (Table 1). Two compounds (i.e., isoamyl alcohol and 2-phenylethanol), excluding the pyrazines, were produced by all bacteria and absent from the TSB media, and were detected with the SPME technique. Isoamyl alcohol was the major compound (i.e., highest relative proportions) detected in all the bacteria via SPME. 

Several species-specific compounds were also detected solely by one method. For instance, 2-phenylethanol was detected only from CF via Tenax^®^ extracts, whereas this compound was identified in all SPME samples (Table 1). Similarly, phenol was found only in CF with SPME. In contrast, indole was detected from all bacteria and TSB except CF, with both methods. Furthermore, all bacteria, apart from CF, produced 2-nonanone, detected using SPME. Some esters, such as ethyl octanoate and hexyl hexanoate, were detected via Tenax^®^ headspace extracts only, and they were unique to EC. Unlike the SPME fiber, the Tenax^®^ filters facilitated the detection of volatile compounds such as 2-coumaranone and o-cymene in CF and KO, respectively. Reciprocally, both isobutyl-alcohol and 3-hydroxy-2-butanone were detected in small quantities from PA by SPME only. 

#### 3.1.2. pH Variation

To further characterize how the chemical composition of bacteria MVOC profiles changed over time with fermentation, we assessed the impacts of pH variations with SPME. Overall, a significant effect of pH was observed on the production of MVOCs across all bacteria (F = 5.06, df = 4,73, *p* = 0.0012; Figure 1). Moreover, the bacteria strain also had significant effects on the production of MVOCs at various pH (F = 3.5, df= 4, 41, *p* = 0.024). While 2-phenylethanol was produced in larger amounts by KO and PA, indole was more abundant in KO extracts when compared to EC, and phenol was only produced by CF (Figure 1). In parallel, NMDS and ANOSIM revealed that various compounds, especially carboxylic acids, were significantly more associated with certain pH (Table 2). For instance, hexanoic acid was more produced at pH 3 (ANOSIM; R = 0.134, *p* = 0.016; Table 2), while pentanoic acid was produced at pH = 7 (ANOSIM; R = 0.125, *p* = 0.022; Table 2).

#### 3.1.3. Inoculated Fruits

Variations in MVOC profiles were also measured from different inoculated fruits. Overall, 42.33% of the variation in the chemical composition is explained by the chosen potential explanatory variables (i.e., bacteria strain, collection methods, type of fruits, and post-inoculation time). MVOCs were detected in apples, kiwifruit, and oranges at 24 h post-inoculation, using both Tenax^®^ and SPME. Multivariate analysis revealed a clear separation between Tenax^®^ and SPME extracts collected at 24 h post-inoculation in apples, kiwifruit, and oranges (ANOSIM; R = 0.19, *p* = 2e-04; Figure 2). A total of seven compounds were significantly associated with Tenax^®^ (i.e., two alcohols, one aldehyde, one aromatic, two hydrocarbons, and two terpenes; Table 3). In line with the previously mentioned variations associated with detection methods, eight compounds, which included indole and 2-phenylethanol, were significantly associated with SPME (Table 3). 

A clear separation was also highlighted between the MVOC profiles of the different types of fruits (ANOSIM; R = 0.5, *p* = 1e-04; Figure 2). Chemical profiles from apples were more similar to kiwifruit, while oranges displayed a different MVOC profile (Figure 2). A total of 40 compounds appeared to explain the difference in odor profiles between the different types of fruits (Table 2). While the chemical profiles of apples were mainly dominated by esters and alcohol, kiwifruit were associated with aromatics, esters and hydrocarbons, and oranges with esters and terpenes (Figure 3; Table 3). It should be noted that nonanal and ethyl butanoate were specific to the chemical profile of kiwifruit, while 2-phenylethanol and ethyl octanoate were specific to oranges (Table 3). In contrast, no compound was significantly associated with any of the bacteria strains inoculated on fruit media (*p* > 0.05).

Moreover, changes in chemical profiles were also investigated over time, by collecting MVOCs in kiwifruit at 24 h, 72 h, and 5 days post-inoculation, using Tenax^®^. Variations in the overall chemical composition of kiwifruit samples were assessed using NMDS and ANOSIM, revealing a clear separation between incubation times (ANOSIM; R = 0.385, *p* = 0.007). A positive correlation was found between the proportions of esters emitted from kiwifruit and the incubation time of bacteria. In contrast, as the number of esters increased over time, fewer hydrocarbons were detected (Figure 3). Furthermore, a total of six compounds were specific to 72 h post-inoculation, while only D-limonene was significantly more associated with 24 h post-inoculation (Table 3). Among the four bioactive compounds identified via SPME-GC/EAD, indole was the one most strongly associated with both the SPME collection method and kiwifruit at 5 days post-inoculation (Table 3). However, no differences in the MVOC profile of kiwifruit were detected between the four bacteria strains (ANOSIM; R = 0.08464, *p* > 0.05).

### 3.2. Electrophysiological Responses 

#### 3.2.1. EAG 

Olfactory responses of virgin females to whole bacterial Tenax^®^ extracts were measured through EAG. The response amplitude varied slightly among bacterial strains (F = 2.45, df = 4, 90, *p* = 0.053), while concentration did not have any effect (F = 0.51, df = 2, 90, *p* = 0.60), nor their interaction (F = 1.49, df = 8, 90, *p* = 0.17). Nonetheless, KO was found to trigger significantly stronger antennal responses than the TSB control (t = 2.73, *p* = 0.008 and t = 1.98, *p* = 0.05, respectively, Figure 4).

#### 3.2.2. SPME-GC/EAD 

Several compounds (Table 1 and Appendix A), among which 2-phenylethanol, indole, isoamyl alcohol, phenol, and pyrazine 2-ethyl-5methyl generated consistent electrophysiological responses from virgin QFF females at the five different pH concentrations (Figure 5). The antennal responses to each of these bioactive compounds were different (F = 3.57, df = 15, *p* < 0.001). However, the variation in the amplitude of antennal responses for each of these compounds at various pH was not consistent with no significant effect of pH (F = 1.194, df = 4, 111, *p* = 0.32) nor in interaction with each compound (F = 0.7, df = 12, 111, *p* = 0.74). 

### 3.3. Behavioral Responses

#### 3.3.1. Cage Assays

When virgin females were offered to choose between bacterial extracts and a control odor (i.e., hexane), a low number (<25%) of flies made a choice within 24 h (Figure 6). Among the proportion of females that made a choice, no preference was found between the bacterial extracts or the control (Chi-square = 77.57, *p* = 0.08). Nonetheless, females displayed a preference towards KO bacteria when compared to CF (z = −3.524, *p* = 0.0024).

#### 3.3.2. Y-Tube Olfactometers

Overall, significantly more females made a choice between either EAD-active synthetic compounds and the control solvent compared to no-choice (Chi-square = 58.504, *p* < 0.001). Among the females that made a choice, virgin females preferred the control compared to 2-ethyl-3-methyl pyrazine (*p* = 0.003), 2-methyl pyrazine (*p* < 0.001), isoamyl alcohol (*p* = 0.047), nonanal (*p* < 0.001), and phenol (*p* < 0.001). In contrast, virgin females preferred 2-nonanone (*p* < 0.001) and D-limonene (*p* = 0.046) over the control (Figure 7). 

## 4. Discussion

Here, we present original results of the MVOCs emitted by Enterobacteriaceae commonly associated with QFF and provide evidence of olfactory and behavioral modulation in virgin females. 

While previous MVOC-related studies generally used SPME fibers and were carried out with bacteria associated with *A. ludens* and/or *Rhagoletis pomonella* [18,26,30,31,32,33], we used and compared the efficiency of two different headspace collection methods, SPME and Tenax^®^. Given that both methods are composed of different adsorbent materials, their efficiency at collecting MVOCs was complementary and used for different bioassays (i.e., solvent Tenax^®^ extracts for behavioral assays and SPME extracts for electrophysiology). In line with previous studies, Tenax^®^ filters adsorbed a larger range of compounds and exhibited a higher sensitivity, while SPME was more efficient at trapping short-chain hydrocarbons [34,35]. Given that certain compounds (e.g., short-chain and volatile molecules) may have had a breakthrough volume based on our Tenax^®^ collection parameters, and because some MVOCs (e.g., 2,5-dimethyl-pyrazine and 3ethyl-2,5-dimethyl-pyrazine) were identified by both methods, while others were detected by only one method, our study highlights the importance of using and comparing various headspace collection methods when investigating MVOCs.

Overall, the chemical composition of the bacterial extracts revealed the presence, in high abundance, of various aldehydes, alcohols, esters, terpenes, and pyrazines, as previously shown [36,37]. Terpenes, esters, aldehydes, and alcohols are the most common chemical classes identified among the wide array of plant volatiles [38] and food lures used against Tephritid flies and detected by the fly antennae [39]. Pyrazines, which are attractive to *A. ludens* and produced by the bacteria *K. pneumonia* and CF [18], were also found, among amides, in the male rectal gland of *B. cucurbitae*, as well as in orchids attractive to males [40]. Although amides were suggested to be involved in the response to sex-specific pheromones in QFF [41], the role of pyrazines remains unclear. In this study, while bacteria were not necessarily responsible for the production of all the detected pyrazines, a wider diversity of these compounds was found in bacteria, when compared to the control media, supporting the findings that bacteria can synthesize certain pyrazines [42].

While most compounds were found across the four studied bacteria, which advocates for the existence of common biosynthetic pathways across Enterobacteriaceae, our chemical analyses also revealed species-specific MVOC production. Although it is still unknown whether Tephritidae flies seek particular bacteria species or strains based on specific MVOC profiles, females were more attracted to indole and 2-phenylethanol, which were detected in all bacteria but CF, than phenol, which was identified in CF extracts only. We showed that the production of MVOCs also depended on the pH of the culture medium, as previously reported [43,44]. For instance, phenol was more abundant in acidic growth media, while KO produced more 2-phenylethanol and indole when cultured in neutral and alkaline pH. Similarly, we reported that chemical profiles were specific to the nature of the growing medium. For instance, esters prevailed in apples, while hydrocarbons and terpenes were mostly present in kiwis and oranges, respectively. While we did not find strong microbially induced specific fruit volatile profiles, the overall abundance of volatiles (i.e., esters, terpenes, alcohol, and aromatics, which were previously described in other fruits commonly attractive to Tephritidae fruit flies [45,46,47]) was enhanced with incubation time with Enterobactereriacea. For instance, a positive correlation was found between the proportion of compounds emitted by kiwifruit and the incubation time (e.g., more D-limonene was produced at 24 h post-inoculation). Given that such fruit volatiles were previously shown to be behaviorally attractive to QFF [6,48,49], we can suppose that the presence of Enterobacteriaceae participates in the fruit attraction. Altogether, our results corroborate the findings that chemical profiles of MVOCs are influenced by the presence of different sources of precursors in the culture media and pH [18,22,25,37,50,51], and they could provide an explanation for the polyphagous capacity of female QFF to exploit several host-plants and adapt to various environmental conditions.

Because female fruit flies may exhibit a delay in sexual maturation and reduced sexual performance and lifespan if they are unable to acquire sufficient protein [10], virgin females are expected to actively search for bacteria as a source of protein. In our study, females exhibited the greatest antennal responses to extracts of KO and EC, which is in line with a previous study showing that KO was the most attractive among several culturable bacterial symbionts of *B. zonata* [52]. Moreover, KO extracts attracted relatively more QFF females compared to CF, similarly to *B. cucurbitae* females, for which KO was more attractive than CF in field experiments [44]. It is noteworthy that CF was previously shown to produce large amounts of ammonia [22], a compound negatively correlated with QFF attraction to *Providencia rettgeri* [51] and other fly species [53,54]. We suggest that the weak attraction to CF in our study may be due to the lack of detection of this compound, as a result of extreme volatility, such as demonstrated in other fruit fly species [55]. While ammonia could not be measured by GC/MS in our study, we acknowledge the existence of kits that enable easy, fast, and reliable detection of ammonia production (e.g., Ammonia Assay Kit, Megazyme, Neogen^®^), and could help understand whether ammonia might be involved in the differential behavioral activity observed between CF and KO.

Among the most abundant volatiles detected in our bacterial extracts, several alcohols (2-phenylethanol, isoamyl alcohol, and phenol), one aromatic (indole), and various pyrazines, triggered antennal responses from virgin females. No differences in the number of attracted *A. ludens* between the solvent and 2-phenylethanol, isoamyl alcohol (syn. 3-methyl butanol) or phenol were previously found [22], while, in another study, negative correlations were found between the production of indole, 2,5-dimethylpyrazine, and 2-phenylethanol by EC, and fly attraction [56]. In contrast, we found that the proportion of EAD-active indole increased with longer incubation times in kiwifruit. While mature fruits were previously described as the most prone to risk of oviposition from fruit flies [57], we suggest that mature fruits may also be at risk from immature fruit flies searching for protein source. From our Y-tube choice assays with synthetic compounds, we found neither a strong attraction nor repulsion to indole nor 2-phenylethanol (at the concentration tested), but phenol, as well as isoamyl alcohol, 2-methyl pyrazine, 2-ethyl 3-methyl pyrazine, and nonanal, appeared to be repellent to virgin female QFF. Moreover, the monoterpene D-limonene, which was the second most abundant compound detected in EC- and KO-inoculated fruits, triggered a strong attraction of virgin females. While the most attractive bacterium in our cage assays (i.e., KO) seemed to produce a repellent compound (i.e., isoamyl alcohol), this may be explained by the fact that behavioral assays were conducted with bacterial Tenax^®^ extracts, which did not contain isoamyl alcohol, although isoamyl alcohol was detected in KO through SPME.

The low degree of behavioral response observed in the cages, reflected by the high proportion of flies not choosing between the bacterial extracts and the control within 24 h, may be explained by the origin of the bacterial strains or the pH at which those extracts were collected. Even though the Enterobacteriaceae species used in this study were previously described to be present in QFF [13,16], they were not sourced directly from the QFF’s gut, which could have influenced the production of MVOCs and, consequently, behavioral attraction [56]. Furthermore, the lack of behavioral attraction to the bacterial extracts may be linked to the origin of the tested flies, which were factory mass-reared and fed with a standard diet based on yeast extract rather than bacteria [58]. As such, behavioral assays with natural QFF populations and screening several strains of live bacteria should be considered.

Most bacteria display optimum growth rates at slightly acidic-neutral pH, but Enterobacteriacea are also found in the midgut of *B. tryoni*, at a pH of 1.4–2.0 [59,60]. During oviposition, the transfer of gut bacteria along eggs as previously described, from a confined acidic environment to a more neutral external environment (i.e., oviposition sites such as fruits [13,61], may trigger an increase in MVOC production. Thus, we suggest that the presence of naturally occurring KO on host-plants, associated with higher release rates of 2-phenylethanol and indole at alkaline pH, may be linked to greater attractiveness to female QFF. Similar findings were reported in *C. capitata*, where an elevation of pH, from acidic to neutral, was correlated with increased release rates of ammonia, resulting in higher bait efficiency [53,62]. Changes in the production of MVOCs by the bacteria transferred to the oviposition site, may provide a new source of chemical information (e.g., timing of oviposition) [13,63], potentially affecting female attraction. Yet, even though female QFF have been observed to reuse previous oviposition sites [64,65], there is still no evidence of MVOCs used as markers of time and quality of oviposition site by this species. Given that only virgin females were used in our study, we cannot confirm that the variation in MVOCs with pH conditions play a specific role in oviposition site selection by mated females. However, future oviposition assays using the MVOCs and bacteria tested in this study should be considered. We also strongly advocate for further comparisons of behavioral modulation between virgin and mated females as a result of mating-induced switches in olfactory preferences previously demonstrated in flies [6,66]. For instance, small alcohol molecules, such as isoamyl alcohol, were recently found to enhance the capture of mated female QFF in the field [46], while we described this compound as repulsive to virgin females.

## 5. Conclusions

By using two different headspace collection methods, we were able to show that the composition of MVOCs produced by bacteria varies according to the bacterial species, the type of media (artificial TSB vs. natural fruit), the pH, and incubation time. Here, we highlighted that the modulation of QFF electrophysiology (EAG) and behavior (cage assay) resulted from a complex interplay between changes in the relative proportions of MVOCs associated with pH, media, and incubation time. Through our results from GC/EAD and Y-tube olfactometer assays, we identified 2-nonanone and D-limonene, and phenol, isoamyl alcohol, and nonanal as candidate compounds for future “attractive” and “repulsive” lures, respectively, for virgin QFF. The elaboration of new female synthetic lures based on these MVOCs will be further tested in the field with natural QFF populations. Our work provides a better understanding of the chemical ecology of fruit flies, which is key to designing novel behaviorally disruptive tools to control QFF.

## Figures and Tables

**Figure 1 microorganisms-11-01643-f001:**
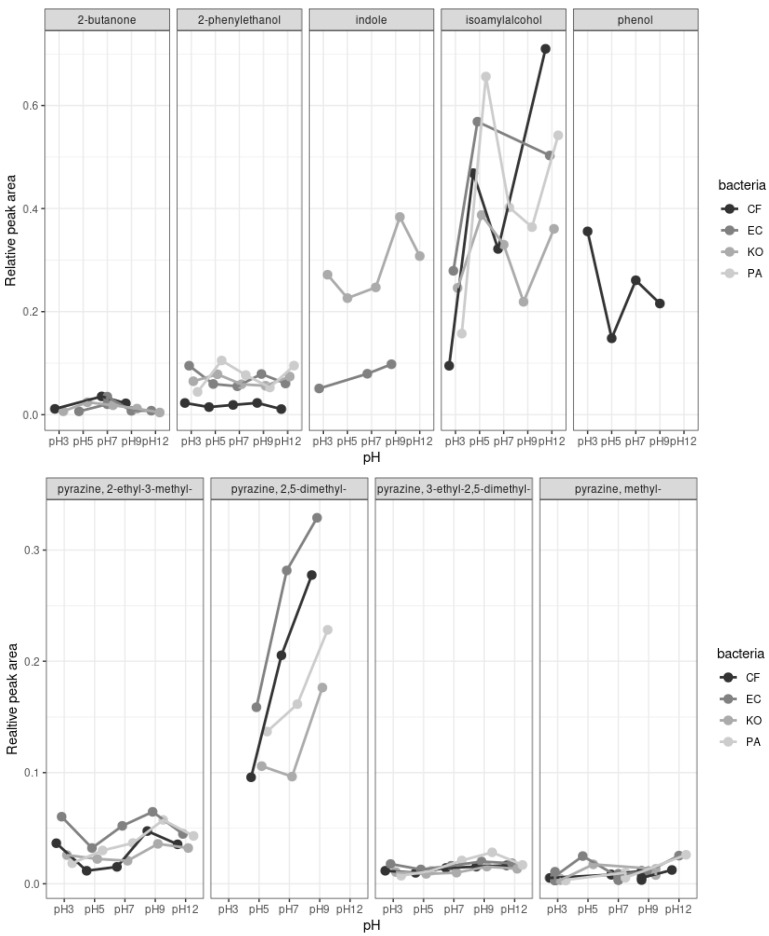
Relative peak area (i.e., total area under the compound peak over the total peak areas from all compounds within a sample) of a selection of antennally active compounds produced by four bacteria and collected by solid phase micro-extraction (SPME). CF, *Citrobacter freundii;* EC, *Enterobacter cloacae*; KO, *Klebsiella oxytoca*; PA, *Enterobacter* (syn. *Pantoea*) *agglomerans*.

**Figure 2 microorganisms-11-01643-f002:**
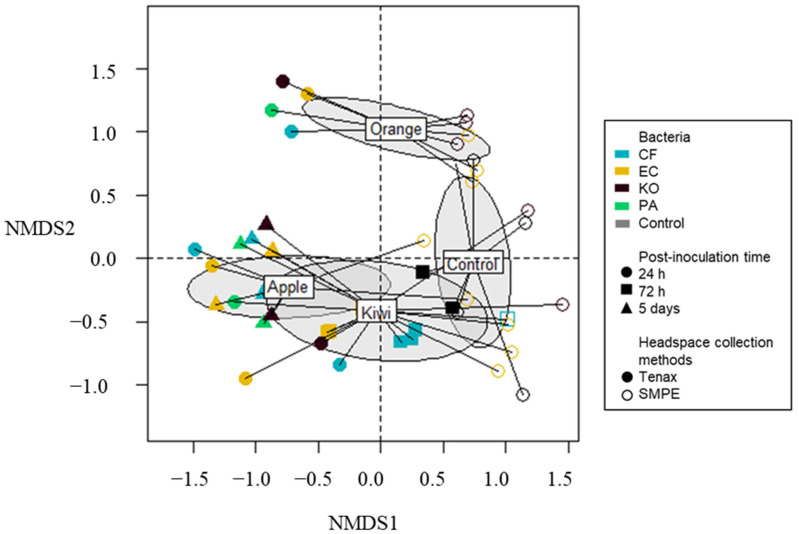
Nonmetric multidimensional scaling (NMDS) plot (stress = 0.107) of the chemical composition of three fruits (apples, kiwifruit, and oranges) and un-inoculated fruits (control). Each point represents a Tenax^®^ (solid)- or SPME-extract sample. The color of the points encodes for the bacterial strains, and the shape encodes for the incubation time prior odor collection. CF, *Citrobacter freundii*; EC, *Enterobacter cloacae*; KO, *Klebsiella oxytoca*; PA, *Enterobacter* (syn. *Pantoea*) *agglomerans*. Centroids of the clusters of each fruit are represented by ellipses (standard deviation in gray). Volatile profiles are significantly different between fruits (ANOSIM, R = 0.5032, *p* = 1e-04), and the collection methods (ANOSIM; R = 0.19, *p* = 2e-04).

**Figure 3 microorganisms-11-01643-f003:**
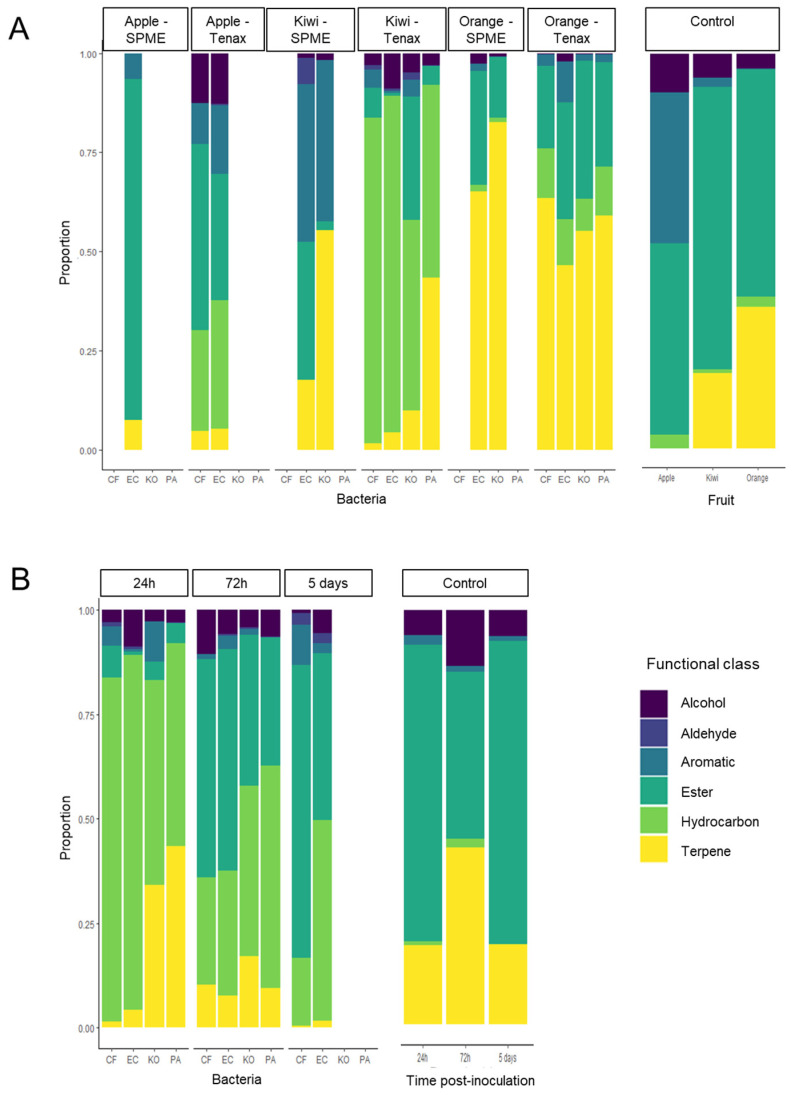
Relative abundance of key chemical classes in fruit samples inoculated by four bacteria strains: CF, *Citrobacter freundii*; EC, *Enterobacter cloacae*; KO, *Klebsiella oxytoca*; PA, *Enterobacter* (syn. *Pantoea*) *agglomerans*. Proportions are represented in apples, kiwifruits, and oranges, collected either via Tenax^®^ or solid phase micro-extraction (SPME) and un-inoculated fruits (control, SPME and Tenax^®^ combined) at 24 h post-inoculation (**A**), and in inoculated kiwifruits and control, collected via Tenax^®^, at 24 h (CF, EC, KO, PA), 72 h (CF, EC), and 5 days (CF, EC, KO, PA) post-inoculation (**B**).

**Figure 4 microorganisms-11-01643-f004:**
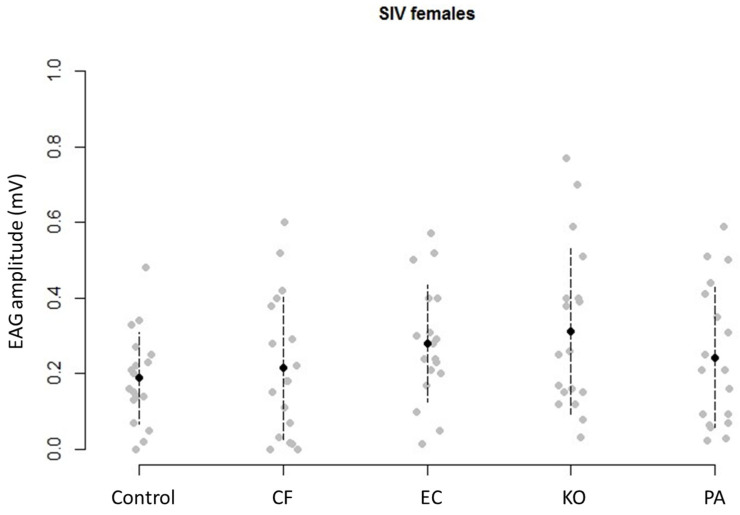
Intensity of electroantennogram (EAG) responses (mV) from virgin female *Bactrocera tryoni* to Tenax^®^ extracts from each bacterium (Control: tryptone soya broth (TSB) media; CF: *Citrobacter freundii*; EC: *Enterobacter cloacae*; KO: *Klebsiella oxytoca*; PA: *Enterobacter* (syn. *Pantoea*) *agglomerans*). EAG responses of all replicates (grey) and average response (black) are shown.

**Figure 5 microorganisms-11-01643-f005:**
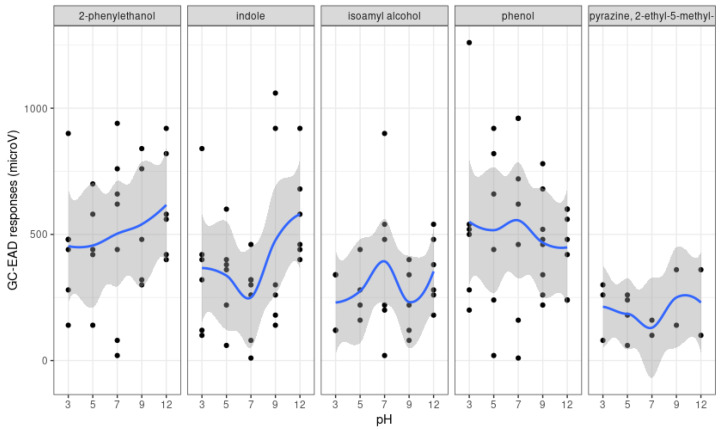
Gas-chromatograph coupled with electroantennographic detection (GC-EAD) responses (in µV) of irradiated virgin female *Bactrocera tryoni* to specific microbial volatile organic compounds produced by a mix of four bacteria at different pH levels collected by solid phase micro-extraction (SPME). Dots represent individual points, and blue lines with greyish areas represent overall mean and standard deviation.

**Figure 6 microorganisms-11-01643-f006:**
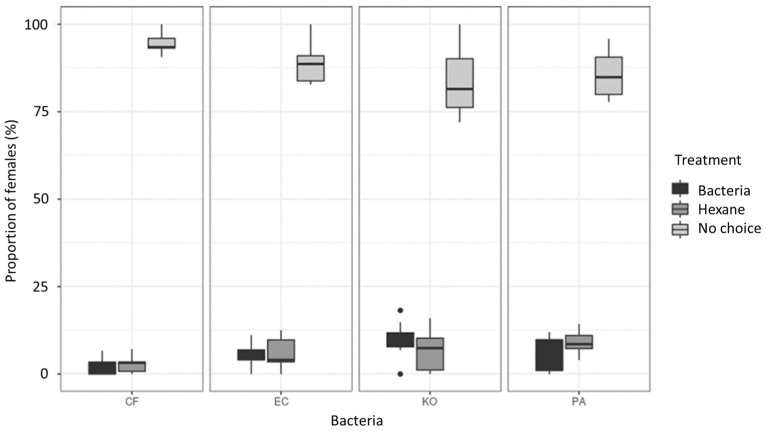
Boxplots showing the percentage in each cage of sexually immature females *Bactrocera tryoni* choosing between the bacterial Tenax^®^ extracts, the control solvent or no-choice after 24 h. Four bacterial extracts were tested as follows: CF, *Citrobacter freundii*; EC, *Enterobacter cloacae*; KO, *Klebsiella oxytoca*; PA, *Enterobacter (syn. Pantoea) agglomerans*. (n = 10 for each bacterium). Black dots are outliers.

**Figure 7 microorganisms-11-01643-f007:**
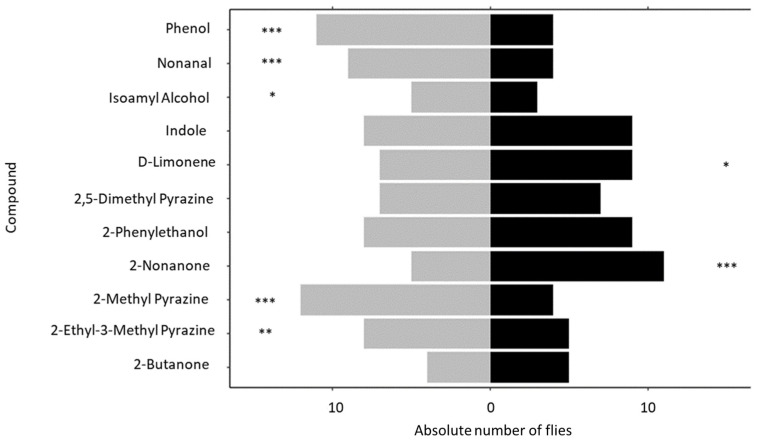
Absolute number of virgin females *Bactrocera tryoni* choosing between individual compounds (black bars) versus the control solvent (grey bars) in a Y-tube olfactometer. (n = 20 for each compound). Statistical significance: *p* ≤ 0.05 (*), *p* ≤ 0.01 (**), *p* ≤ 0.001 (***).

**Table 1 microorganisms-11-01643-t001:** Mean relative proportions of microbial volatile organic compounds (MVOCs) identified for each bacterium grown at pH 7 collected by 24 h headspace sample with either Tenax^®^ filter or solid phase micro-extraction (SPME) fiber, followed by solvent extraction. The MVOCs in bold letters were identified to trigger SPME-Gas-chromatograph coupled with electroantennographic detection (GC/EAD) responses, and MVOCs highlighted in grey were tested in Y-tube olfactometer with 11 synthetic compounds. (CF: *Citrobacter freundii*; EC: *Enterobacter cloacae;* KO: *Klebsiella oxytoca*; PA: *Enterobacter* (syn. *Pantoea*) *agglomerans* and tryptone soya broth (TSB) media). * KI: Kovats retention index; ** RT: retention time (in min).

	Mean RT ** (min.)		Mean KI *		CF		EC		KO		PA		TSB	
Compounds	Tenax	SPME	Tenax	SPME	Tenax	SPME	Tenax	SPME	Tenax	SPME	Tenax	SPME	Tenax	SPME
2-Butanone		3.35		599		0.04		0.03		0.03				
Isobutyl alcohol		3.86		638								0.01		
3-Hydroxy- 2-butanone, (syn. acetoin)		4.84		704								0.06		
**Isoamyl alcohol**		**5.48**		**731**		**0.36**		**0.3**		**0.46**		**0.48**		
Methyl-pyrazine		7.85		823		0.01		0.01				0.01		0.04
1,2-Dimethyl-benzene	8.7		<900				0.01		0.01					
Isoamyl acetate (syn. 1-Butanol, 3-methyl-, acetate)		9.57		874				0.02		0.1		0.03		
Nonane	9.6		902				0.01		0.01		0.01			
2,5-Dimethyl-pyrazine	9.9	10.82	909	911	0.26	0.23	0.05	0.36	0.15	0.14	0.04	0.19	0.05	
2,6-Dimethyl-pyrazine	9.9	10.83	911	912			0.07				0.37			0.64
Benzaldehyde	11.7		960		0.04		0.02		0.05		0.05		0.14	
**Phenol**		**13.43**		**979**		**0.29**								
2-Ethyl-3-methyl-pyrazine		14.25		1001		0.02		0.07		0.03		0.04		0.12
2-Ethyl-5-methyl-pyrazine	13.1	14.31	1000	1003	0.04	0.02	0.02		0.02		0.03			
O-Cymene	14.1		1026						0.01					
D-Limonene	14.3		1030				0.12		0.13					
Phenylacetaldehyde	14.8		1044				0.01		0.01		0.02			
Ɣ-Terpinene	15.4		1060				0.02		0.03					
1-Octanol	15.8		1072				0.01		0.01					
3-Ethyl-2,5-dimethyl-pyrazine	16.1	17.11	1079	1075	0.09	0.02	0.07	0.02	0.04	0.01	0.07	0.03	0.06	0.06
2-Nonanone		17.7		1090		0		0.02		0.01		0.04		
Dimethyl-2-vinylpyrazine	16.9		1099		0.03		0.04		0.02		0.03			
Linalool	16.9		1101				0.02		0.02					
Nonanal	17.1		1105		0.03		0.04		0.02		0.01		0.05	
**2-Phenylethanol**	**17.4**	**18.23**	**1113**	**1104**	**0.03**	**0.02**		**0.07**		**0.04**		**0.09**		
Ethyl Octanoate	20.5		1198				0.01							
Dodecane	20.6		1200		0.06		0.1		0.07		0.05		0.11	
Decanal	20.8		1206				0.02		0.01		0.01			
3-Phenyl-furan	21.3	22.64	1222	1222				0.01		0.01	0.01	0.02		0.03
(E)-2-Dodecenal	21.6		1229				0.01							
2-Coumaranone	21.7		1233		0.04									
Neral	22		1242				0.03		0.02					
Citral	23.1		1272				0.05		0.04					
**Indole**	**23.8**	**25.18**	**1294**	**1294**				**0.1**		**0.17**	**0.16**		**0.15**	**0.02**
2,5-Dimethyl-3-(3-methylbutyl)-pyrazine		25.77		1311								0.01		0.02
Hexyl hexanoate	26.9		1387				0.01							
Tetradecane	27.4		1400		0.31		0.25		0.22		0.14		0.31	
Dodecanal	27.7		1409				0.01							
1-Dodecanol	29.7		1474		0.07				0.1				0.13	

**Table 2 microorganisms-11-01643-t002:** List of individual compounds that are significantly associated with the different types of absorbent (i.e., Tenax^®^ filter and solid phase micro-extraction; SPME fiber), strains of bacteria (CF, *Citrobacter freundii*; EC, *Enterobacter cloacae*; KO, *Klebsiella oxytoca*; PA, *Enterobacter* (syn. *Pantoea*) *agglomerans*), and pH. Both ANOSIM R value and *p*-values are reported for each compound.

Group: SMPE					Group: Bacteria Mix + PA			
Compound	Functional Class	ANOSIM Statistics			Compound	Functional Class	ANOSIM Statistics	
		R Value	*p*-Value				R Value	*p*-Value
2-Ethyl-3-methyl-pyrazine	Aromatic	0.739	0.0082		2-Phenylethanol	Alcohol	0.128	0.0426
** Group: Tenax **					** Group: KO + PA **			
Compound	Functional class	ANOSIM statistics			Compound	Functional class	ANOSIM statistics	
		R value	*p*-value				R value	*p*-value
Benzaldehyde	Aldehyde	0.715	0.0082		Decane, 5-methyl-	Hydrocarbon	0.129	0.0461
3-Ethyl-2,5-dimethyl-pyrazine	Aromatic	0.750	0.0227		** Group: PA + TSB **			
Dimethyl-2-vinylpyrazine	Aromatic	0.781	0.0488		Compound	Functional class	ANOSIM statistics	
Nonanal	Aldehyde	0.832	0.0082				R value	*p*-value
Dodecane	Hydrocarbon	0.922	0.0082		Phenylacetone	Ketone	0.128	0.0411
Tetradecane	Hydrocarbon	0.939	0.0082		** Group: bacteria mix + EC + KO **			
					Compound	Functional class	ANOSIM statistics	
** Group: bacteria mix **							R value	*p*-value
Compound	Functional class	ANOSIM statistics			Pyrazine, 2-ethyl-3,5-dimethyl-	Aromatic	0.141	0.0177
		R value	*p*-value		** Group: KO + PA + TSB **			
Pentadecane	Hydrocarbon	0.160	0.0057		Compound	Functional class	ANOSIM statistics	
1-Butanamine, 3-methyl-	Amine	0.149	0.024				R value	*p*-value
Nonane, 2,5-dimethyl-	Hydrocarbon	0.147	0.0248		Hexadecane	Hydrocarbon	0.13	0.037
** Group: CF **								
Compound	Functional class	ANOSIM statistics			** Group: pH3 **			
		R value	*p*-value		Compound	Functional class	ANOSIM statistics	
Nonane, 4,5-dimethyl-	Hydrocarbon	0.168	0.0068				R value	*p*-value
** Group: EC **					Hexanoic acid	Carboxylic acid	0.134	0.0164
Compound	Functional class	ANOSIM statistics			** Group: pH5 **			
		R value	*p*-value		Compound	Functional class	ANOSIM statistics	
Benzene, 1,2,4-trimethyl-	Terpene	0.165	0.001				R value	*p*-value
** Group: KO **					3-Methyl-2-thiophenecarboxaldehyde	Aldehyde	0.112	0.0324
Compound	Functional class	ANOSIM statistics			** Group: pH7 **			
		R value	*p*-value		Compound	Functional class	ANOSIM statistics	
Benzeneacetaldehyde	Aldehyde	0.153	0.0021				R value	*p*-value
Acetic acid	Carboxylic acid	0.133	0.0424		Pentanoic acid	Carboxylic acid	0.125	0.0215
** Group: PA **					** Group: pH5 + pH12 **			
Compound	Functional class	ANOSIM statistics			Compound	Functional class	ANOSIM statistics	
		R value	*p*-value				R value	*p*-value
2-Dodecanone	Ketone	0.155	0.0115		Pyrazine, 2,5-dimethyl-3-(3-methylbutyl)-	Aromatic	0.142	0.004
** Group: TSB **					Propanoic acid	Carboxylic acid	0.134	0.0093
Compound	Functional class	ANOSIM statistics			p-cresol	Aromatic	0.131	0.0115
		R value	*p*-value					
Cyclohexanone, 5-methyl-2-(1-methylethyl)-, trans-	Ketone	0.152	0.0108					
** Group: bacteria mix + CF **								
Compound	Functional class	ANOSIM statistics						
		R value	*p*-value					
Nonanal	Aldehyde	0.15	0.0121					

**Table 3 microorganisms-11-01643-t003:** List of individual compounds that are significantly associated with different types of fruits: apple, kiwifruit, and oranges (ANOSIM, R = 0.5032, *p* = 1e-04); collection methods: Tenax^®^ filter and SPME fiber (ANOSIM; R = 0.19, *p* = 2e-04); and incubation times: 24 h, 72 h, and 5 days post-inoculation (ANOSIM; R = 0.385, *p* = 0.007). Statistical significance: *p* ≤ 0.05 (*), *p* ≤ 0.01 (**), *p* ≤ 0.001 (***).

Group: SMPE					Group: Apple			
Compound	Functional Class	ANOSIM Statistics			Compound	Functional Class	ANOSIM Statistics	
		R Value	*p*-Value				R Value	*p*-Value
Indole	Aromatic	0.499	0.0088 **		Hexyl benzoate	Ester	0.789	0.0001 ***
β-Pinene	Terpene	0.479	0.0147 *		Estragole	Aromatic	0.695	0.0001 ***
Phenylethyl alcohol	Alcohol	0.448	0.0063 **		2,3-Dimethyldecane	Hydrocarbon	0.613	0.0006 ***
Caryophyllene	Terpene	0.446	0.0155 *		Hexyl 2-methylbutyrate	Ester	0.606	0.0002 ***
D-Limonene	Terpene	0.436	0.0362 *		α-Farnesene	Terpene	0.603	0.0001 ***
Methyl octanoate	Ester	0.43	0.0274 *		Hexyl acetate	Ester	0.597	0.0002 ***
Naphtalene	Hydrocarbon	0.427	0.0155 *		Benzyl acetate	Ester	0.519	0.0003 ***
Ethyl octanoate	Ester	0.418	0.0292 *		Propylheptanol	Alcohol	0.426	0.0414 *
** Group: Tenax **					(Z)-4-Decen-1-ol	Alcohol	0.423	0.0414 *
Compound	Functional class	ANOSIM statistics			Butyloctanol	Alcohol	0.417	0.0414 *
		R value	*p*-value		2-Hexyl-1-decanol	Alcohol	0.41	0.0414 *
(E)-2-Nonen-1-ol	Alcohol	0.598	0.0003 ***		Butyl caprylate	Ester	0.406	0.0414 *
2,2,4-Trimethyl-1,3-pentanediol diisobutyrate	Hydrocarbon	0.566	0.0001 ***		Ethyl (4E)-4-decenoate	Ester	0.405	0.0414 *
Valencen	Terpene	0.432	0.0056 **		Butyl Acetate	Ester	0.354	0.0287 *
β-Selinene	Terpene	0.431	0.0372 *		** Group: Kiwifruit **			
Hexane, 2,4-dimethyl-	Aldehyde	0.421	0.0044 **		Compound	Functional class	ANOSIM statistics	
2-Hexyldecanol	Alcohol	0.406	0.0132 *				R value	*p*-value
m-Xylene	Aromatic	0.376	0.0412 *		Nonanal	Aldehyde	0.56	0.0022 **
					Ethyl butanoate	Ester	0.398	0.0478 *
** Group: Orange **								
Compound	Functional class	ANOSIM statistics			** Group: 24h post-inoculation **			
		R value	*p*-value		Compound	Functional class	ANOSIM statistics	
Ethyl octanoate	Ester	0.747	0.0001 ***				R value	*p*-value
Methyl octanoate	Ester	0.715	0.0001 ***		D-Limonene	Terpene	0.693	0.0273 *
Caryophyllene	Terpene	0.67	0.0001 ***		** Group: 72h post-inoculation **			
Ethyl hexanoate	Ester	0.666	0.0002 ***		Compound	Functional class	ANOSIM statistics	
β-Pinene	Terpene	0.662	0.0001 ***				R value	*p*-value
D-Limonene	Terpene	0.609	0.0005 ***		Indole	Aromatic	0.623	0.0121 *
Phenylethyl alcohol	Alcohol	0.566	0.0014 **		** Group: 5 days post-inoculation **			
Eugenol	Aromatic	0.559	0.0005 ***		Compound	Functional class	ANOSIM statistics	
Naphtalene	Hydrocarbon	0.555	0.0002 ***				R value	*p*-value
Limonene oxide, trans-	Aromatic	0.544	0.0025 **		β-Elemen	Terpene	0.989	0.0009 ***
Ethyl hydroxy-3-hexanoate	Ester	0.528	0.0025 **		α-Maaliene	Terpene	0.951	0.0009 ***
Butyl hexanoate	Ester	0.522	0.0008 ***		2-Hexyl-1-octanol	Alcohol	0.93	0.0025 **
Hexyl hexanoate	Ester	0.508	0.0004 ***		2,3-Dimethyldecane	Hydrocarbon	0.89	0.0030 **
α-Maaliene	Terpene	0.507	0.0041 **		β-Selinene	Terpene	0.886	0.0010 ***
β-Selinene	Terpene	0.506	0.0034 **		Ethyl benzoate	Ester	0.801	0.0044 **
Ethyl decanoate	Ester	0.503	0.0025 **					
Valencen	Terpene	0.498	0.0044 **					
Ethyl dodecanoate	Ester	0.492	0.0025 **					
E)-4,8-Dimethylnona-1,3,7-triene	Hydrocarbon	0.49	0.0025 **					
E-Nerolidol	Alcohol	0.479	0.0081 **					
3-Carene	Terpene	0.469	0.0087 **					
Hexyl octanoate	Ester	0.426	0.0143 *					
Butyl butanoate	Ester	0.391	0.0129 *					
β-Elemen	Terpene	0.381	0.0185 *					

## Data Availability

The data presented in this study are available within the article supplementary materials. All R scripts used in for all data analyses are available on request from the corresponding author.

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
