# Peer review of "Electrophysiological and Behavioral Responses of Virgin Female Bactrocera tryoni to Microbial Volatiles from Enterobacteriaceae"

_microorganisms, 2023, doi:10.3390/microorganisms11071643_

Round 1

Reviewer 1 Report

The  paper is well-written and reports some interesting, albeit not hugely significant data on virgin female Bactrocera tryoni responses to microbial volatiles. The authors correctly identify limitations such as the origin of the tested bacteria not being from the fruit fly itself (and consequently the term 'symbiont' may not be appropriate). The behavioural responses to isolated volatiles are not dramatic and this begs the question of whether analysing responses to cocktails of volatiles might be more revealing.

Issues to respond to:

Lines 347-357: the findings reported here suggest lack of quantitative recovery of volatiles dependent on the head-space method employed. The authors claim the methods are complimentary, but an important caveat concerns their reproducibility. It is not clear to this referee why one method captures a specific volatile from one source but not from another.

Line 434: 'Eelectrohysiological'

Discussion: this is currently too long and should be shortened to minimise duplication of the results sections, especially where reported differences in volatile compositions are not huge, especially bearing in mind technical issues concerning their recovery. 

The authors cite a paper from 1985 that indicates the fly gut pH is between 1.4 and 2. This seems extraordinarily low and not an environment where the specific bacteria will succeed. How does this pH value compare to that of other fruit flies? Maybe the gut pH needs to be revisited.

Author Response

Response to reviewer 1

The paper is well-written and reports some interesting, albeit not hugely significant data on virgin female Bactrocera tryoni responses to microbial volatiles. The authors correctly identify limitations such as the origin of the tested bacteria not being from the fruit fly itself (and consequently the term 'symbiont' may not be appropriate). The behavioural responses to isolated volatiles are not dramatic and this begs the question of whether analysing responses to cocktails of volatiles might be more revealing.

We thank the reviewer for their analysis of our manuscript.

Issues to respond to:

Lines 347-357: the findings reported here suggest lack of quantitative recovery of volatiles dependent on the head-space method employed. The authors claim the methods are complimentary, but an important caveat concerns their reproducibility. It is not clear to this referee why one method captures a specific volatile from one source but not from another.

We thank the reviewer for their concerns. As mentioned in the manuscript and reported in previous studies, we suggest that both methods should be employed in a complementary manner. Conventional headspace collection methods (e.g., Tenax) generally exhibit higher sensitivity for highly volatile compounds, whereas SPME headspace technique was found to be more adapted to the collection of less volatile compounds. Given that in the case of absorption/solvent injections (e.g., Tenax), the solvent peak can sometimes overlap with low molecular weight and highly volatile compounds with low retention times, the use of solvent-free methods, such as SPME, is particularly relevant for the collection microbe-produced volatiles. Yet, a lot of options regarding the model of SPME fiber are now available. For instance, among convenient fibers, PDMS fiber was shown to be less suited for quantitative analyses of certain types of compounds as its coating tends to discriminate against polar and highly volatile compounds (Pelusio et al., 1995). This is also why we wanted to couple the use of SPME technique with Tenax extracts to get a comprehensive picture of chemical profiles in our study. 

Nonetheless, it is not always clear when one sampling method should be preferred over another, as previously suggested (Rering 2020a, Pelusio et al., 1995, Cognat 2012, Krumbein and Ulrich., 1996). Indeed, qualitative, and quantitative differences in chemical profiles may not necessarily be due to design-related trapping efficiency or chemical bias of the adsorbent material, but also to the sampling itself, and be strongly influenced by the type of medium (or odor source) investigated, we want to emphasize the importance of considering the strengths and weaknesses of several headspace collection methods for exploratory approaches. For instance, headspace Tenax was better suited for quantitative analyses of volatile organic sulfur compounds found in truffle aromas (Pelusio et al., 1995), while both Tenax and SPME methods gave comparable results when examining tomato aroma (Krumbein and Ulrich., 1996).

Overall, SPME should be preferred for straightforward analysis of major volatile components, while Tenax sampling methods may be one of the methods of choice for trace analysis.

Regarding the reviewer’s concern of data reproducibility, we compared the mean coefficient of variance (VC) for all the identified peaks between both methods. The VC, which is a statistical measure of the relative dispersion of data points in a data series around the mean, was calculated by dividing the standard deviation by the mean and multiplied by 100 (see table below). The mean CV for all of the compounds present in Tenax extracts was 74.8% and 64.4% for all the peaks identified via SPME, and no differences in repeatability was found for the two techniques (p-value = 0.3573).

Compound

SPME

Tenax

Mean

VC

VC(%)

Mean

VC

VC(%)

(E)-2-Nonen-1-ol

NA

NA

NA

2E+05

2E+05

75.746

(Z)-4-Decen-1-ol

NA

NA

NA

2E+05

28742

16.943

.m-Xylene

NA

NA

NA

2E+05

1E+05

83.005

2,2,4-Trimethyl-1,3-pentanediol diisobutyrate

NA

NA

NA

1E+07

1E+07

115.7

2,3-Dimethyldecane

NA

NA

NA

3E+05

2E+05

57.282

2,6-Dimethyl-7-octen-2-ol

NA

NA

NA

4E+05

3E+05

86.081

2-Butyloctanol

NA

NA

NA

1E+06

9E+05

87.774

2-Hexyl-1-decanol

NA

NA

NA

6E+05

2E+05

38.914

2-Hexyl-1-octanol

NA

NA

NA

6E+05

3E+05

46.247

2-Hexyldecanol

NA

NA

NA

9E+05

9E+05

103.4

3-Carene

NA

NA

NA

2E+06

1E+06

67.775

9-Decene-1-nitrile

NA

NA

NA

2E+05

51441

23.674

Benzyl acetate

NA

NA

NA

2E+05

2E+05

96.998

Butyl Acetate

6E+06

6E+06

93

NA

NA

NA

Butyl (E)-2-hexenoate

NA

NA

NA

3E+05

39020

14.713

Butyl butanoate

NA

NA

NA

4E+05

5E+05

108.01

Butyl caprylate

NA

NA

NA

2E+05

70516

43.107

Butyl hexanoate

2E+07

1E+07

89.41

2E+06

2E+06

133.33

Butyloctanol

NA

NA

NA

1E+05

29556

28.385

Caryophyllene

3E+06

2E+06

50.16

3E+05

1E+05

41.673

Caryophyllene oxide

NA

NA

NA

3E+05

NA

NA

D-Limonene

4E+08

4E+08

112.8

1E+07

2E+07

148.01

E)-4,8-Dimethylnona-1,3,7-triene

NA

NA

NA

2E+06

1E+06

57.138

E-Nerolidol

NA

NA

NA

8E+05

5E+05

62.831

Estragole

NA

NA

NA

6E+05

3E+05

58.388

Ethyl (4E)-4-decenoate 

NA

NA

NA

2E+05

92672

44.374

Ethyl benzoate

NA

NA

NA

6E+05

5E+05

83.822

Ethyl butanoate

8E+06

7E+06

92.44

9E+06

1E+07

141.43

Ethyl decanoate

NA

NA

NA

2E+05

1E+05

50.055

Ethyl dodecanoate

NA

NA

NA

4E+05

2E+05

55.729

Ethyl heptanoate

NA

NA

NA

2E+05

17466

10.702

Ethyl hexanoate

1E+07

7E+06

63.37

3E+06

3E+06

91.079

Ethyl hydroxy-3-hexanoate

NA

NA

NA

3E+05

86461

34.488

Ethyl octanoate

4E+07

3E+07

75.64

5E+06

7E+06

144.03

Eugenol

2E+06

6E+05

25.15

2E+06

2E+06

133.87

Hexane, 2,4-dimethyl-

NA

NA

NA

2E+05

2E+05

124.12

Hexyl 2-methylbutyrate

NA

NA

NA

1E+05

82393

65.406

Hexyl acetate

NA

NA

NA

8E+05

7E+05

81.44

Hexyl benzoate

NA

NA

NA

93121

23561

25.301

Hexyl hexanoate

2E+07

2E+07

89.97

1E+06

3E+06

193.37

Hexyl octanoate

NA

NA

NA

1E+06

1E+06

89.687

Indole

4E+06

2E+06

52.13

1E+06

2E+06

138.3

Isoamyl acetate

NA

NA

NA

50904

NA

NA

Limonene oxide, trans-

NA

NA

NA

2E+06

3E+05

21.129

Linalool

NA

NA

NA

3E+05

1E+05

37.388

Methyleugenol

8E+05

6E+05

75.94

1E+06

6E+05

48.873

Methyl hexanoate

NA

NA

NA

85967

73496

85.493

Methyl octanoate

7E+06

3E+06

47.08

1E+06

2E+05

14.723

Naphtalene

6E+06

5E+06

85.71

NA

NA

NA

Nonanal

9E+05

4E+05

44.98

1E+06

5E+05

53.093

Phenylethyl alcohol

6E+06

5E+06

87.2

NA

NA

NA

Propylheptanol

NA

NA

NA

2E+05

5877

3.3644

Pyrazine,2,5-dimethyl

1E+06

0

0

NA

NA

NA

Valencen

NA

NA

NA

7E+06

9E+06

131.66

α-Farnesene

7E+05

89113

12.46

2E+06

2E+06

111.44

α-Maaliene

NA

NA

NA

7E+05

8E+05

107.05

β-Elemen

NA

NA

NA

6E+05

8E+05

141.87

β-Phenethyl acetate

NA

NA

NA

77144

NA

NA

β-Pinene

2E+07

1E+07

61.74

1E+06

3E+05

22.911

β-Selinene

NA

NA

NA

3E+05

2E+05

84.243

Line 434: 'Eelectrohysiological'

Amended.

Discussion: this is currently too long and should be shortened to minimise duplication of the results sections, especially where reported differences in volatile compositions are not huge, especially bearing in mind technical issues concerning their recovery. 

Thank you for this comment. As a result, we have shortened the discussion, from 2,006 words (first version of the manuscript) to 1,521 words.

The authors cite a paper from 1985 that indicates the fly gut pH is between 1.4 and 2. This seems extraordinarily low and not an environment where the specific bacteria will succeed. How does this pH value compare to that of other fruit flies? Maybe the gut pH needs to be revisited.

Several previous publications have described that certain region of the midgut of female fruit flies may be extremely acidic (less than 3), such as in Drosophila melanogaster (Veenstra, J.A., et al. 2008). Moreover, similar pH levels to those reported in B. tryoni (Drew et al. 1983) were also found depending on the region and bacterial strains. For instance, while a highly alkaline ventriculus and a neutral-to-acidic hindgut were described in D. melanogaster, an acidic crop was found in response to the passage of Firmicutes and Proteobacteria within the alimentary tract (Thaochan, et al., 2010). Overall, previous studies have suggested different optimum pH for bacteria depending on the specific activities in various gut regions (Andres, et al., 2007).

References

Fabio Pelusio, Torben Nilsson, Luca Montanarella, Roberto Tilio, Bo Larsen, Sergio Facchetti, and Jorgen 0. Madsen,  (1995), Headspace Solid-Phase Microextraction Analysis of Volatile Organic Sulfur Compounds in Black and White Truffle Aroma, Jour. of Agric. Food Chem, 43, 2138-2143, https://pubs.acs.org/doi/pdf/10.1021/jf00056a034?casa_token=nyf4xLLBg58AAAAA:gwlZEpNtbZLq-r_sD8DDM7zPyZjvMDqN5aj8XXD-xl4Z7RCNxHJVOBP9PIwGhjcZrJxY-N12VFK-UutVjw

Claudine Cognat, Tom Shepherd, Susan R. Verrall, Derek Stewart, (2012), Comparison of two headspace sampling techniques for the analysis of off-flavour volatiles from oat-based products, Food Chemistry, 134, 3, 1592-1600, https://doi.org/10.1016/j.foodchem.2012.02.119

Krumbein and Ulrich., (1996), Comparison of three sample preparation techniques for the determination of fresh tomato aroma volatiles, Falvour Science: Recent Developments, 291-296, Elsevier, ISBN: 9781845698232

Thaochan et al., (2010), Alimentary tract bacteria isolated and identified with API-20E and molecular cloning techniques from Australian tropical fruit flies, Bactrocera cacuminataand B. tryoniJournal of Insect Science, 10, 1, 131, https://doi.org/10.1673/031.010.13101

Andres, V.S., Ortego, F. and Castañera, P. (2007), Effects of gamma-irradiation on midgut proteolytic activity of the mediterranean fruit fly, Ceratitis capitata (Diptera: Tephritidae). Arch. Insect Biochem. Physiol., 65: 11-19. https://doi.org/10.1002/arch.20172

Caitlin C. Rering, Jose G. Franco, Kathleen M. Yeater, Rachel E. Mallinger, (2020), Drought stress alters floral volatiles and reduces floral rewards, pollinator activity, and seed set in a global plant, Ecosphere, 11, 9, https://doi.org/10.1002/ecs2.3254

Veenstra, J.A., Agricola, HJ. & Sellami, A. Regulatory peptides in fruit fly midgut. Cell Tissue Res 334, 499–516 (2008). As reported in other studies, the pH in the middle midgut can be extremely low  https://doi.org/10.1007/s00441-008-0708-3 https://link.springer.com/article/10.1007/s00441-008-0708-3

Reviewer 2 Report

The manuscript by Tallon et al. describes the identification of volatiles emitted by different enterobacteriaceae grown on artificial medium or on host fruits of the queensland fruitfly. Two different odor collection methods were compared and both electroantennogram and behavioral responses of virgin females to individual compounds were analyzed. The paper shows differences in the identification of compounds between SPME and Tenax collections as expected. Some differences in volatiles emitted by different bacteria species were identified and some influence of pH on volatile emission was also revealed. When growing bacteria on host fruits, differences in volatile emissions were essentially due to the type of fruit, whereas no compound was significantly associated with any of the bacteria species used for inoculation.

Then electroantennographic responses to various compounds eluting from the GC were recorded and antennal responses to different bacterial extracts. Finally behavioural experiments showed no significant choice of bacterial strain extracts as compared to the rearing medium and Y-tube olfactometer analyses showed attraction to two previously identified compounds and a repulsive effect of five compounds. The results are interesting, but due to the restriction on virgin females somewhat preliminary, as mentioned in the discussion of the manuscript. I encourage the authors to provide some more details on the methods as indicated below and showing some more details of the results would help to clarify the below mentioned points.

Part 2.6.

The method description is very brief for the EAG: either refer to another paper for details, or add information on the stimulation device, amplifier, acquisition and software.

For GC-EAD recording, did you really use Autospike software? Not rather GC-EAD software by Syntech?

Part 2.7.

Description of Y-tube olfactometer experiments: It is confusing that you talk first about 30 min experiment duration and then say that flies not having made a choice by the end of the 15 minutes were recorded as no choice. Which time is correct?

Part 3.1.3.

Comparison of inoculated fruit: Maybe I have missed something, but isn’t there a control missing: un-inoculated fruits? As you say, many of the compounds you identified are also present in plants…

Figure 4: I would entitle the y-axis “EAG amplitude (mV)”, “intensity” is misleading

GC-EAD results, Fig. 5: This part needs clarification. I wonder, if it would not be useful to show GC-EAD trace examples, with indication of the compounds, which you show in Fig. 5?

Do you want to say that, because different quantities of the shown compounds are emitted at different pH, the EAG response is accordingly more or less strong? Can you refer to Figure 1? But response amplitudes are different for different compounds: it is for example interesting to see that isoamyl alcohol is highly present in SPME collections, but elicits relatively small EAD responses. Inversely 2-phenylethanol is present in a small amount, but elicits strong EAD responses.

Part 3.3.2.

I am not sure that I understand the sense of the first sentence. For some compounds insects don’t choose, for others they do in one direction or the other, but does the global analysis in this case add any useful information?

Line 153: … of each of the three replicates.

Line 171: …Tenax extracts and SPME samples

Line 178: …maintained at 250°C

Line 192/193: …exported under quarantine to New Zealand…

Line 210: …responses of virgin females

Line 223: …tested in concentrated form

Line 228-239: please correct the temperature values for °C (not oC)

Line271: …one of the end arms,…

Table 2 and 3: Group SPME (not SMPE)

Line 360: …over time with….

Line 384: …two alcohols…

Author Response

Response to reviewer 2

The manuscript by Tallon et al. describes the identification of volatiles emitted by different enterobacteriaceae grown on artificial medium or on host fruits of the Queensland fruit fly. Two different odor collection methods were compared and both electroantennogram and behavioral responses of virgin females to individual compounds were analyzed. The paper shows differences in the identification of compounds between SPME and Tenax collections as expected. Some differences in volatiles emitted by different bacteria species were identified and some influence of pH on volatile emission was also revealed. When growing bacteria on host fruits, differences in volatile emissions were essentially due to the type of fruit, whereas no compound was significantly associated with any of the bacteria species used for inoculation.

Then electroantennographic responses to various compounds eluting from the GC were recorded and antennal responses to different bacterial extracts. Finally behavioural experiments showed no significant choice of bacterial strain extracts as compared to the rearing medium and Y-tube olfactometer analyses showed attraction to two previously identified compounds and a repulsive effect of five compounds. The results are interesting, but due to the restriction on virgin females somewhat preliminary, as mentioned in the discussion of the manuscript. I encourage the authors to provide some more details on the methods as indicated below and showing some more details of the results would help to clarify the below mentioned points.

We thank the reviewer for their analysis of our manuscript.

Part 2.6.

The method description is very brief for the EAG: either refer to another paper for details, or add information on the stimulation device, amplifier, acquisition and software.

Amended.

For GC-EAD recording, did you really use Autospike software? Not rather GC-EAD software by Syntech?

Autospike software from Syntech was used for analyzing results from the SPME-GC-EAD recordings.

Part 2.7.

Description of Y-tube olfactometer experiments: It is confusing that you talk first about 30 min experiment duration and then say that flies not having made a choice by the end of the 15 minutes were recorded as no choice. Which time is correct?

The flies were given 15 minutes to make a choice between the different types of odorant stimuli. We have adjusted the description in the methods to avoid confusion.

Part 3.1.3.

Comparison of inoculated fruit: Maybe I have missed something, but isn’t there a control missing: un-inoculated fruits? As you say, many of the compounds you identified are also present in plants…

The reviewer is correct. We had control fruits which were inoculated with 10μl of distilled water only (description in the methods, line 119 and highlighted in pink in Supplementary table 3). For more transparency, we added these data to Figure 2 and Figure 3. However, in table 3, only individual compounds that are significantly associated with different conditions (i.e., here, types of fruits, collection methods and incubation times) are shown. As no compound was significantly associated with control fruits (un-inoculated), this condition is not shown in this table.

Figure 4: I would entitle the y-axis “EAG amplitude (mV)”, “intensity” is misleading.

Amended.

GC-EAD results, Fig. 5: This part needs clarification. I wonder, if it would not be useful to show GC-EAD trace examples, with indication of the compounds, which you show in Fig. 5?

Do you want to say that, because different quantities of the shown compounds are emitted at different pH, the EAG response is accordingly more or less strong? Can you refer to Figure 1? But response amplitudes are different for different compounds: it is for example interesting to see that isoamyl alcohol is highly present in SPME collections, but elicits relatively small EAD responses. Inversely 2-phenylethanol is present in a small amount, but elicits strong EAD responses.

Yes, the reviewer interpreted it correctly. Because the measurements of GC-EAD were done using volatiles collected via SPME and from a mix of all bacteria, which were adjusted to the different pH (3,5,7,9 and 12), we decided to present figure 5 with the x-axis as pH. Therefore, the reviewer is correct when saying that these results should be explained in the context of changes of volatiles produced by each bacterium at different pH (shown in Figure 1). These results are now emphasized in the shortened discussion (i.e., small EAD responses to isoamyl alcohol, which is highly abundant in SPME collections, in contrast to strong EAD responses to 2-phenylethanol, which was detected in small proportions). In case this can help the reviewer, below are supplementary figures, showing changes in FID quantity for each compound-of-interest produced by the mix bacteria at the different pH. We did not deem necessary to include these figures in the manuscript.

Part 3.3.2.

I am not sure that I understand the sense of the first sentence. For some compounds insects don’t choose, for others they do in one direction or the other, but does the global analysis in this case add any useful information?

We have modified the first sentence. Among all tested flies, significantly more insects made a choice between the solvent (control) and the MVOC-of-interest (odorant stimulus), when compared to flies that did not choose any side of the Y-tube olfactometer, which highlights the reliability of our assay. If most insects had not made a choice between both types of odorant stimuli, it would be difficult to infer any activity (attraction or repulsion) to the tested compounds.

All comments below were amended. We thank the reviewer for their comments.

Line 153: … of each of the three replicates.

Line 171: …Tenax extracts and SPME samples

Line 178: …maintained at 250°C

Line 192/193: …exported under quarantine to New Zealand…

Line 210: …responses of virgin females

Line 223: …tested in concentrated form

Line 228-239: please correct the temperature values for °C (not oC)

Line271: …one of the end arms,…

Table 2 and 3: Group SPME (not SMPE)

Line 360: …over time with….

Line 384: …two alcohols…

Reviewer 3 Report

The article submitted for review concerns the physiological and behavioural response of fruit flies to bacterial-derived volatiles. This issue is important for both cognitive and practical reasons. The knowledge gained can, in fact, be used to develop new methods to control the abundance of Bactrocera tryoni populations.

The article was well written and carefully prepared. The introduction outlines the problem well. The results are described in a concise but comprehensive manner. The discussion is extensive and the conclusions are supported by the results. The article should be accepted for print after minor corrections have been made.

Minor comments

L69: B tryoni – insert dot

L99 and 117: verify the number

L116: verify the UV spectrum

Check and correct the degree sign throughout the manuscript.

Unify P value throughout the manuscript. I suggest abandoning the exponential notation.

Author Response

We thank the reviewer for their comments. All their suggestions were taken into consideration while reformating our manuscript. 

Round 2

Reviewer 2 Report

Thank you for the careful revision of the manuscript, which addresses all points I evoked. There are a few typing and language errors remaining, I  recommend to check the manuscript carefully during editing/proof reading.

Author Response

Dear reviewer,

We thank you for your attention to our manuscript. Typos and errors were corrected, and all references standardized. A final proofreading was conducted. Following the journal editor's and your recommendations, we are pleased to resubmit our manuscript.

On behalf of all the co-authors,

Sincerely,

Anais K. Tallon
